## RESEARCH ARTICLE

# Estimating the impact of Viamo call-in information services on sexual and reproductive health knowledge, behavior, and outcomes among women of reproductive age: A 2-arm open label randomized controlled trial in Uganda

Günther Fink[1,2]*, Branwen Nia Owen[1,2], Jan Will[3], Ilyena Kozain[4]

1 University of Basel, Basel, Switzerland, 2 Swiss Tropical and Public Health Institute, Allschwil, Switzerland, 3 Innovations for Poverty Action, Kampala, Uganda, 4 Viamo, Kampala, Uganda

* guenther.fink@swisstph.ch

## Abstract

### Background

Despite substantial increases in access to schooling and to the internet, health information and knowledge remains limited among women of reproductive health in many low- and middle-income countries. Given the near universal cell-phone coverage in low- and middle-income countries, call-in services offer an interesting new platform to provide high quality information to targeted populations. In this paper, we assessed the impact of the Viamo 3-2-1 call-in platform on health knowledge and behavior among women of reproductive age in Uganda.

### Methods

We conducted a 2-arm open label randomized controlled trial targeting 6000 women, ages 18–49 years, with access to a simple feature Airtel cell phone in Uganda. Fifty enumeration areas were randomly selected in Kampala, Madi-Okollo, Rwampara and Katakwi districts for a total of 200 enumeration areas. Consenting women were asked to complete a baseline and endline survey implemented by trained interviewers using tablets through home visits. After completion of baseline, women were randomized with equal probability to treatment and control using a tablet-generated random number draw. Study staff introduced treated women to the call-in service and encouraged its use through various incentives and promotional messages. Primary outcomes were sexual and reproductive health knowledge scores as well as iron supplementation in pregnancy. Secondary outcomes included self-reported use of contraception, healthy diets, birth spacing and birth outcomes as well as hemoglobin levels and BMI. Linear regression models were used to estimate mean differences in outcomes between treatment and control. Standard errors were corrected for the cluster-based sampling of women.

**Data availability statement:** We have uploaded a replication package file which contains all data as well as the code used to create figures and tables.

**Funding:** This project was funded through USAID DIV grant number 7200AA21FA00018. The funders had no role in study design, data collection and analysis, decision to publish, or preparation of the manuscript.

**Competing interests:** The first, second and third author declare no competing interest. The last author (IK) is currently employed by Viamo, the company operating the service analyzed. She was not involved in the data analysis and did not modify the way the results are presented or described in any way.

## Results

A total of 6,011 women were enrolled into the study between December 6, 2022 and February 11, 2023; 3,052 were assigned to treatment, 2,959 used as controls. End-line data was collected between May 8, 2024 and August 4, 2024. Overall follow-up rates were 92% in both the treatment and the control group. Treatment increased average sexual and reproductive health knowledge by 0.07 standard deviations (95% CI [0.01, 0.11], p-value = 0.02) and increased the likelihood of taking iron supplements by 4% pts among recently pregnant women ([0.003, 0.08], p-value 0.04). Results for secondary outcomes were mixed, with positive changes for nutritional intake and family planning behaviors and intentions and no impact on hemoglobin levels and BMI. No adverse events or outcomes were reported.

## Conclusions

The results of this trial suggest that access to call-in services can increase health knowledge and to some extent also health behaviors. Longer and more intensive program exposure may be needed to yield measurable changes in reproductive outcomes and nutritional status.

## Introduction

In low- and middle-income countries (LMIC), access to timely and accurate information in national languages is often limited, particularly among the poor. Whereas the internet is a common source of information in many countries, only 10% of Ugandans, and 36% of Sub-Saharan Africans were estimated to use the internet in 2021; far lower than the global average of 63% [1]. This lack of information access can contribute to poor health decisions; this is particularly important for women of reproductive age, since lack of information often leads to inadequate prenatal care [2] and ineffective family planning methods [3], threatening not only the life of mothers [4] but also their children [5,6].

Public health messaging in LMICs is often disseminated through mass media channels such as radio messages and billboards. These channels have proven to be effective at reaching large numbers of people [7] and increasing knowledge, but have had an insufficient impact on changing behaviors in many areas of public health [8,9]. Mass media campaigns often fail to create behavior change because, in part, people cannot access the information at a moment of need.

The rapid growth in cell phone coverage in recent years [10] offers new opportunities to address this gap, as information can be accessed at any time, from anywhere. Recent evidence suggests that interventions to improve reproductive health knowledge through easy access technologies can effectively improve reproductive health knowledge and use of modern contraception [11–13].

In this study, we aimed to assess whether providing access to call-in-services – free numbers that women can dial to obtain information on health – can effectively improve sexual and reproductive health knowledge and attitudes among women of reproductive health in Uganda.

## Methods

### Study design

This study was designed as a two-arm, non-blinded, individual-level randomised controlled trial focusing on women's nutrition and reproductive health. Given that the main intervention – a phone-based, voice information service – was equally available to both arms, we used an encouragement design, actively introducing women in the treated arm to the service and encouraging them to use it throughout the study period. Blinding of the intervention was not possible given the nature of the program.

### Study setting and participant recruitment

This study was conducted in four districts of Uganda: Kampala, Katakwi, Madi-Okollo and Rwampara. These four districts were purposely chosen to cover all major regions of Uganda. To recruit women within these four districts, a multi-stage sampling process was followed: first, a list of all enumeration areas in each district was obtained, and 50 enumeration areas were randomly chosen using a simple random number draw created with the Stata statistical software package. In rural areas, enumeration areas corresponded to villages, and in urban areas, to residential blocks. All households within each selected enumeration area were visited to identify eligible women. Fig 1 shows the spatial location of the study districts within Uganda.

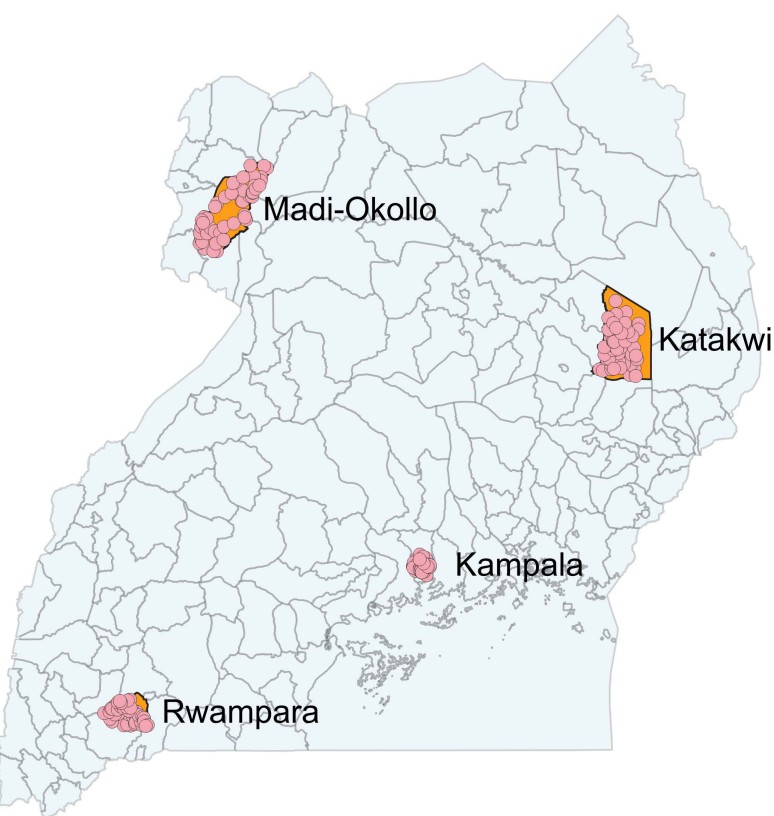

**Fig 1. Spatial distribution of study districts (orange shapes) and study clusters (small circles).** Uganda administrative shapefiles provided by https://gadm.org/data.html. Figure created by authors.

## Inclusion and exclusion criteria

In a second step, study eligibility was assessed by trained study enumerators. Women living in selected enumeration areas were eligible for inclusion in the study if they were aged 18–49 years and had access to an Airtel (a mobile network operator) phone number within their household. Given the explicit focus on antenatal care, family planning and post-partum, all women who were pregnant during the listing were automatically invited to join the study; non-pregnant women were randomly selected for the study until the 45 women per cluster target was reached. In clusters with less than 45 women, all women were invited to participate in the study.

Women were ineligible if they had used the Viamo platform previously, they planned to move outside of the study area within 12 months of baseline or if they did not speak one of the six languages available on the platform: Ateso, English, Luganda, Lugbara, Luo or Runyakitara.

There were no exclusion criteria (or exclusions) during the study.

## Intervention

Viamo's platform was created with the objective of addressing the need for accessible and timely health information. Through an easy to remember dial-in number, callers can obtain up-to-date news as well as key information on topics related to nutrition, maternal and child health, and sexual and reproductive health. The platform is free of charge, and currently has over 4 million users in Uganda, and 35 million users in LMICs globally [14]. Users who call hear a pre-recorded, interactive voice response (IVR) menu, and can choose topics to listen to from these menus.

In Uganda, the Viamo platform is available via voice and text channels on all mobile phones with an Airtel number and offers a wide variety of health and other information to all interested callers.

For the study, we aimed to generate random variation in system usage by providing strong encouragement to use the system to a randomly selected subset of women. Specifically, to ensure increased service utilization among women randomly selected for the treatment, all treated women in the sample received:

1. A personalized introduction and mock call into platform (at time of baseline interview)

2. A promotional calendar to serve as a physical reminder on how to navigate the service and the benefits of the service (at time of baseline interview)

3. Continued encouragement to use the service in the form of text messages and outbound calls, incentivizing women to call the service.

The original sixty sexual and reproductive, maternal and neonatal health and nutrition key messages launched at the inception of the project were developed in conjunction with subject experts, end users, communication specialists, and the Ugandan Ministry of Health (MOH). Throughout the implementation period, the key messages (2–4 mins of dialogue) were transformed into stylized dynamic content (7–45 mins), to reiterate critical concepts for enhanced learning and be able to promote new content. The release of DIV-dynamic content formats (dramas, talk shows, and celebrity interviews) was staggered and began in August 2022.

Treated women were encouraged to use the system whenever they had questions or wanted to know more about sexual and reproductive health. Viamo also employed targeted nudges, personalized campaigns and other outreach activities (such as release of new content), with the aim of maintaining and incentivizing usage rates (active calling into the Viamo Platform) among women in the treatment group (see Supplemental Materials S1 File for an example).

## Study procedures

Study staff visited selected women at their homes, introduced the study and invited them to participate. The consent form was read aloud and explained when necessary. Literate women signed the consent form and illiterate women marked the

form with a thumbprint, and a literate witness was asked to sign on their behalf. We did not assess participants' cognitive capacity to provide consent.

Baseline data was collected via face-to-face interview at each participants' home between December 6, 2022 and March 6, 2023. Upon completion of the baseline survey, treated women were introduced to the service by study staff. Endline data was collected approximately 15 months after recruitment, between May 8, 2024 and August 4, 2024 in two separate surveys; a phone survey and a face-to-face in-person survey (see Supplementary Table 1 for dates by district, and Supplementary Table 2 for data collected by phone and in-person interview at endline). During all three surveys, the interviewer recorded responses on a tablet using the SurveyCTO software. Data collectors received training and on-going support and monitoring to ensure data quality. Participants received UGX10,000 (~USD $2.77) of phone credit as a token of gratitude after each survey interview.

Surveys collected data on socio-demographic characteristics, knowledge and use of the Viamo platform, reproductive health knowledge, contraceptive use, pregnancy planning and uptake of antenatal care, pregnancy outcomes, diet and exercise behaviors and select biomarkers.

### Randomization

Upon completion of the baseline survey, we randomised arm allocation at the individual level based on a simple number draw generated by the SurveyCTO software. Specifically, at the end of the survey, the software generated a random number from a uniform distribution [0,1] and allocated all draws > 0.5 to treatment.

Women selected for the treatment arm were then provided with a personalized introduction to the service (after the baseline survey was completed) and called the service directly from their own phone with the support of the interviewer to see how the system works. Treated women were also given a promotional calendar to serve as a physical reminder on how to navigate the service and the benefits of the service. Women in the control group were not given any intervention. Given that the service was freely available to all women, they were in theory able to call the service at any time, but not given any encouragement or support to do so by the study team.

### Power calculations

The study was powered to detect a 0.1 standard deviation (SD) difference in standardized knowledge scores (mean 0, SD 1) with power 0.9. Assuming an average cluster size of 30, a follow-up rate of 90% and a design effect of 1.3, this required 100 clusters per arm, and a total sample size of 6000 women. The targeted effect size was based on an anticipated knowledge score difference of 0.2 SD among compliers (system users), and an average compliance of 10% in the control, and 60% in the treatment arm (a net compliance/usage differential of 50%).

### Primary and secondary outcomes

The two primary outcomes of the study were reproductive health knowledge and nutritional supplementation in pregnancy. For reproductive health, we used a 15-item questionnaire of true/false statements (items listed in Supplemental Materials S2). This tool was based on previous studies conducted in sub-Saharan Africa [11] and pretested locally to make sure all items were suitable for the study population. To facilitate interpretation of regression results, we converted the raw scores (number of correct answers out of 15) into z-scores with mean zero and standard deviation one. For nutrition knowledge, we analyzed a binary indicator for women (falsely) believing that herbs can cure anemia. For nutritional supplementation during pregnancy, we used self-reported iron supplementation as an outcome measure. Secondary outcomes included contraceptive knowledge (number of methods respondents are familiar with), contraceptive use (currently using any modern method), prenatal care uptake (any, number of visits), hemoglobin level (as a biomarker indicating actual iron intake), body mass index, and birth spacing.

For pregnancy related outcomes, we analyzed two separate sets of women: first, women reporting to be pregnant at the time of the endline survey (currently pregnant), and second, women who completed a pregnancy in the year preceding the endline survey (recently pregnant). For current pregnancies, we analyzed self-report iron supplementation as well as self-reported use of antenatal care services, and whether women indicate the pregnancy was planned. For recently completed pregnancies, we analyzed the same variables, but also analyzed pregnancy loss and low birth weight (reported weight at birth < 2.5kgs) as additional secondary outcomes.

## Statistical analysis

We began our analysis with a comparison of baseline characteristics across arms, as well as a comparison of follow-up rates to assess potential selective attrition. All primary models were estimated intent-to-treat, comparing outcomes between treated and control women at endline. For all outcomes, we estimated both unadjusted linear regression models (not including any covariates) and fully adjusted models, where we controlled for age, district, marital status, educational attainment and baseline knowledge variables. The main rationale for using linear regression models was to generate mean differences estimates that could be compared across the rather large set of outcome variables. To adjust for non-normal residuals, we used Huber's cluster-robust standard errors [15].

We used forest plots to illustrate item level changes for aggregated reproductive health knowledge and contraceptive use scores. In an exploratory analysis, we also estimated per protocol models and stratified models. Per protocol models were restricted to compliant women: specifically, we included only women in the control group who had never called the service, and women in the treatment group who had called the service at least five times. To investigate differential impact on subgroups, we estimated stratified models by age group (<30, 30+), educational attainment group (primary or less, secondary, higher) and district.

## Study registration and ethical clearance

The trial was registered on June 03, 2022 at https://www.socialscienceregistry.org/trials/9544. Ethical clearance for the study was obtained from the Mildmay Uganda Research Ethics Committee (MUREC-2022–124), the ethics committee for North-Western Switzerland (AO_2022–00034) as well as by the Innovations for Poverty Action internal review board (#00006083).

## Results

Of 54,259 women screened, 29% (n = 15,532) were eligible to participate in the study (Fig 2). The most common reason for ineligibility was not having access to an Airtel number. A total of 6,011 women completed the baseline survey and were randomised to the control (n = 3,052) or the intervention arm (n = 2,959). At endline, 87% (n = 2,678) of control arm participants, and 86% (n = 2,555) of intervention arm participants completed the phone survey. Slightly fewer participants completed the in-person survey (81% of both arms). 92% of participants in both arms completed at least one of the two modalities (i.e., either a phone or an in-person survey). The most common reason for loss to follow were participants being unreachable by phone, and participants having moved house, in the phone survey and in-person survey, respectively.

Table 1 shows descriptive statistics for the study sample. Over half were under 30 years old (53%), and a majority were currently married (58%). Twelve percent (12%) of subjects were recruited in Madi-Okollo, 21% in Katakwi, 32% in Rwampara and 35% in Kampala. Fewer than half of women lived in a household with piped water (47%) and very few had health insurance (2%), but most owned their own phone (84%). Most had previously been pregnant (88%) and nearly a seventh were currently pregnant (14%). The trial arms were well-balanced, with all characteristics very similar across arms (Table 1).

Table 2 summarizes overall compliance with treatment and platform use. 50% of women in the control group indicated at endline that they had heard of the Viamo platform compared to 96% in the intervention arm. 20% of women in the

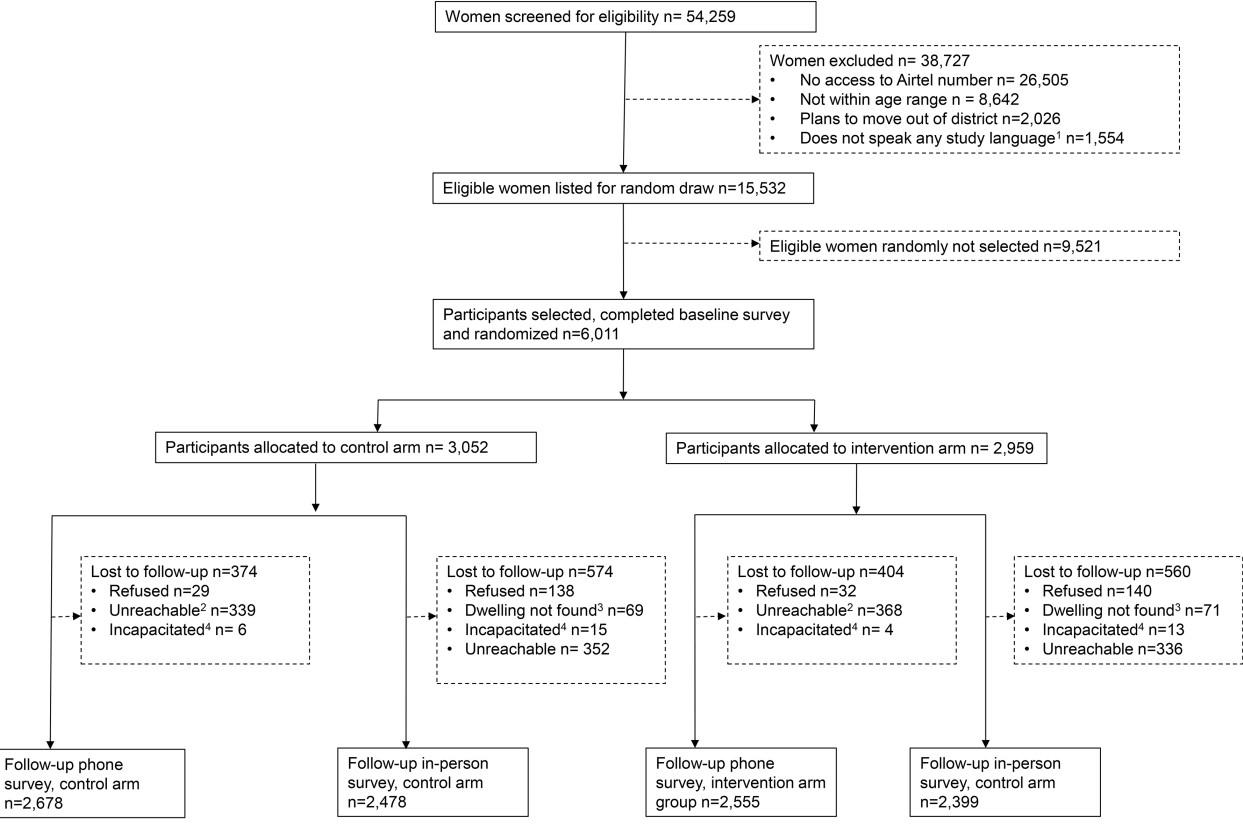

**Fig 2. Consort Trial participant flow chart.** *Notes*: [1] Ateso, English, Luganda, Lugbara, Luo or Runyakitara; [2] Number out of service, phone being switched off, or calls unanswered. [3] Household visited at baseline was not found again. [4] Ill, intoxicated, or deceased.

control arm reported to have ever used the platform, and 7% reported to have used it in the last month. In the treatment arm, the same was true for 84% and 42% respectively. On average, subjects in the control arm reported four calls to the platform, women in the treatment arm reported an additional 18 calls on average. These numbers also align nicely with the Viamo platform metrics, which recorded an average of four calls for the control group, and an additional 31 calls among women in the treatment group (suggesting that on average women under-reported the number of actual calls they made). In a given month, a treatment user listened to an average of approximately 38 minutes of content on the platform.

Table 3 provides an overview over the most popular content. The most popular single item was the local weather forecast, followed by information on food and general news. On average, 71% of the content listened to by women in the treatment group accessing the platform (n = 2,825) was content specifically related to maternal health, sexual and reproductive health, family planning, or nutrition. For the control group members who accessed the platform (n = 1,024) only 20% listened to the same thematically relevant content for this study.

Table 4 shows estimated impact on knowledge. We find a 0.07 SD increase in the primary knowledge outcome score in unadjusted models, and an increase of 0.06 SDs in fully adjusted models. Treatment also increased the average number of reproductive methods women reported to be familiar with by 0.11 points (implying that one in 10 women know one additional method). For nutrition, the treatment reduced the false belief that local herbs could cure anemia by 3% pts.

Fig 3 shows item-specific changes in sexual and reproductive health knowledge. Largest improvements were observed for myths regarding male condoms causing infertility and post-sex urination preventing pregnancy; small improvements were seen for most questions.

**Table 1.  Sample characteristics overall and by study arm.**

|  | Overall (n=6,011) | Control (n=3,052) | Treatment (n=2,959) |
|---|---|---|---|
| Characteristic | N (%) | N (%) | N (%) |
| Age 18–29 | 3,159 (53%) | 1,609 (53%) | 1,550 (52%) |
| Age 30–39 | 1,913 (32%) | 979 (32%) | 934 (32%) |
| Age 40+ | 939 (16%) | 464 (15%) | 475 (16%) |
| Currently married | 3,503 (58%) | 1,756 (58%) | 1,747 (59%) |
| No education | 159 (2.6%) | 95 (3.1%) | 64 (2.2%) |
| Primary education | 3,173 (53%) | 1,640 (54%) | 1,533 (52%) |
| Secondary or higher education | 2,611 (43%) | 1,285 (42%) | 1,326 (45%) |
| Education: other | 68 (1.1%) | 32 (1.0%) | 36 (1.2%) |
| Speaks English | 1,716 (29%) | 862 (28%) | 854 (29%) |
| Kampala | 2,081 (35%) | 1,071 (35%) | 1,010 (34%) |
| Katakwi district | 1,256 (21%) | 619 (20%) | 637 (22%) |
| Madi-Okollo district | 741 (12%) | 385 (13%) | 356 (12%) |
| Rwampara district | 1,933 (32%) | 977 (32%) | 956 (32%) |
| Household has piped water | 2,799 (47%) | 1,431 (47%) | 1,368 (46%) |
| Household has health insurance | 100 (1.7%) | 45 (1.5%) | 55 (1.9%) |
| Woman has her own phone | 5,056 (84%) | 2,527 (83%) | 2,529 (85%) |
| Currently pregnant | 867 (14%) | 441 (14%) | 426 (14%) |
| Previously pregnant (ever) | 5,275 (88%) | 2,683 (88%) | 2,592 (88%) |

**Table 2.  Treatment compliance.**

| Outcome | Heard of the platform | Ever used the platform | Used the platform in the last month | Total number of self-reported calls to platform[a)] | Total number of registered calls to platform |
|---|---|---|---|---|---|
| Treatment effect | 0.459*** | 0.642*** | 0.346*** | 18.25*** | 31.41*** |
|  | (0.0192) | (0.0143) | (0.0129) | (0.471) | (0.958) |
| Control group average | 0.499 | 0.499 | 0.0709 | 4.075 | 4.012 |
| Observations | 5,233 | 5,233 | 5,233 | 5,233 | 5,233 |
| R-squared | 0.262 | 0.412 | 0.164 | 0.268 | 0.271 |

Notes: All point estimates displayed correspond to estimated mean differences in outcomes. Numbers in parentheses are cluster-robust standard errors. Asterisks indicate p-value range of estimated differences: *p-value<0.10, **p-value<0.05, ***p-value<0.01. [a)] Self-reported number of calls were top-coded at 51 calls.

Supplemental Materials Figure SF1 further illustrates the observed changes in knowledge for the item with the largest absolute change in knowledge score. Supplemental Materials Table ST1 shows the proportion of women answering each item correctly at baseline and endline by treatment status for all items investigated.

Table 5 summarizes the estimated impact on sexual behavior as reported in person as well as on the phone. The two survey modalities yielded consistent results with 40% of women using modern contraception and the intervention resulting in an insignificant additional 2% of women using modern contraception. Positive impacts were found for the number of methods ever used (+0.14 methods in adjusted models) and the intention to use modern contraception in the future (+ 5% pts).

Fig 4 provides further details on current use of contraceptive methods with a focus on methods ever used. The intervention increased the use of implants, pills and condoms by 3–4% pts – changes on other methods were not significant.

Table 6 shows estimated impact on prenatal care. A total of 398 current pregnancies, as well as 653 pregnancies completed between baseline and endline were analyzed. No differences were found in antenatal care attendance or

**Table 3. Top ten most popular messages listened to by treatment group on the platform (Dec 2023 – July 2024).**

| No | Message Title | Key Themes | Times Reached | Times Listened | Total Number of Listeners | C2L Ratio |
|---|---|---|---|---|---|---|
| 1 | Dynamic daily weather report | weather | 14140 | 14140 | 2140 | 100% |
| 2 | To learn about what food keeps us healthy | health, nutrition, anemia | 8858 | 6710 | 1753 | 89% |
| 3 | Dynamic daily news bulletin | news | 5959 | 4488 | 1046 | 86% |
| 4 | To know what a healthy weight is | health, nutrition, BMI | 4335 | 3585 | 1310 | 92% |
| 5 | To learn more about cervical cancer | SRH | 4056 | 3417 | 1179 | 91% |
| 6 | What is folic acid and why take it? | health, pregnancy | 4006 | 3177 | 1208 | 89% |
| 7 | What is sex? | puberty, SRH | 3852 | 3500 | 1513 | 95% |
| 8 | DIV Drama Episode 1: "Family planning for a better future" | FP, SRH | 3675 | 2053 | 633 | 77% |
| 9 | To know the four ways to improve your chance for a healthy pregnancy | SRH, pregnancy | 3660 | 2205 | 1453 | 78% |
| 10 | To learn how iron keeps us strong | nutrition, anemia | 3558 | 2950 | 1092 | 89% |

*Notes*: A listener is defined as someone who has listened to at least 75% of a key message. *C2L (caller-to-listener ratio) defines the % completion listening rate of a message, i.e., the proportion of callers who listen to the message all the way to the end. BMI = body mass index, FP = family planning, SRH = sexual and reproductive health

**Table 4. Impact on knowledge.**

| Outcome | Reproductive health knowledge z-score | Beliefs that local herbs cure anemia | Number of contraceptive methods familiar with |
|---|---|---|---|
| Treatment effect | 0.0708*** | −0.0371** | 0.116** |
| (unadjusted) | (0.0265) | (0.0152) | (0.0461) |
| Adjusted treatment effect | 0.0591** | −0.0328** | 0.0941** |
| Control group average | 9.742 | 0.406 | 10.33 |
| Survey modality | In person | In person | Phone |
| Observations | 4,877 | 4,877 | 5,233 |

*Notes*: Table shows estimated impact on primary outcomes. Coefficients displayed are mean differences with standard errors in parentheses. Adjusted models control for age group (age < 20, age 20–34 (ref), age >=35), educational attainment, district, baseline family planning knowledge, baseline hemoglobin level and baseline body mass index. Numbers in parentheses are cluster-robust standard errors. Asterisks indicate p-value range of estimated differences: *p-value<0.10, **p-value<0.05, ***p-value<0.01

deworming. Treatment increased the likelihood of taking iron in current pregnancies by 8% pts and the likelihood of taking supplements in recently completed pregnancies by 4% pts [0.3,7.7].

Table 7 shows estimated impact on other behavior. The intervention increased women's efforts to eat healthy by 4% pts; interestingly, these efforts were generally not linked to losing weight; on average, treated women were 2.5% less likely to attempt for weight loss, and 3% pts more likely to try to gain weight. In terms of food intake, the treatment appears to have increased the frequency of fruit and vegetable consumption, consistent with the results on healthier diets. This aligns with consistent messaging on increasing healthy habits such as regular activity, the inclusion of nutritious and iron rich foods, and maintaining a healthy body weight.

Table 8 shows estimated impacts on pregnancy and birth outcomes. 41% of currently pregnant women in the control group reported that the current pregnancy was not planned; this proportion was 5% pts smaller in the treatment group, but not significantly different. Unplanned pregnancies in the past were slightly more common among treated women (but not statistically different). The proportion of previous pregnancies lost as well as the proportion of most recent births born with low birthweight (< 2500 grams) were substantially lower in the treated group – these differences were however not significant statistically due to the small number of observations here.

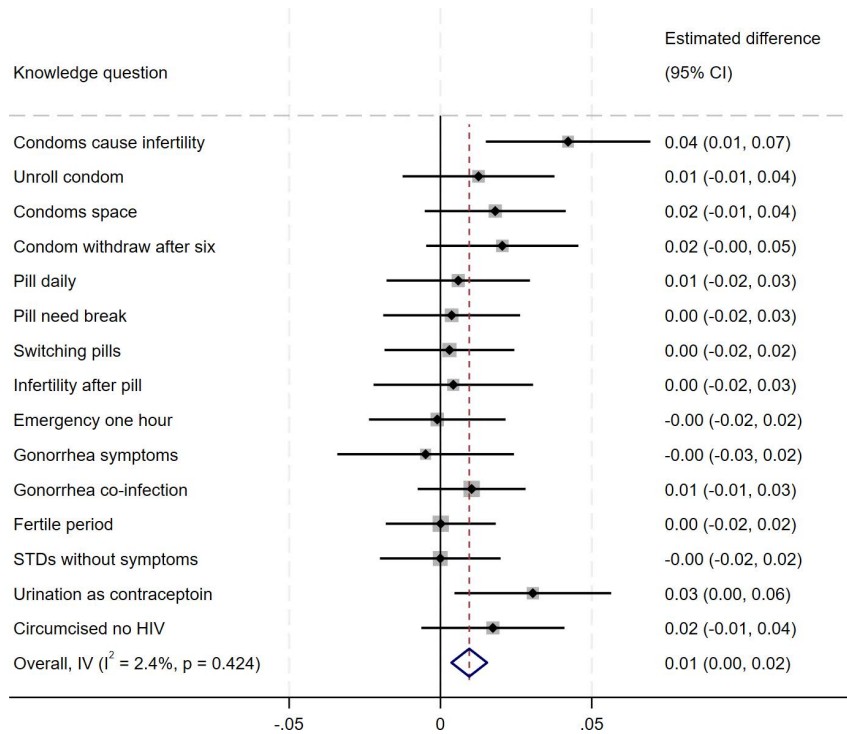

**Fig 3. Item-level change in knowledge.** *Notes:* each point/square on the graphic represents the estimated coefficient obtained from regressing a binary indicator of giving a correct answer to the specific question on treatment – lines illustrate 95% confidence intervals.

**Table 5. Impact on self-reported sexual behavior.**

| Outcome | Currently uses any contraception[a] | Currently uses modern contraception[a] | Can show contraception to interviewer | Number of methods ever used | Currently uses any contraception[a] | Currently uses modern contraception[a] | Plans to use modern contraception in the future |
|---|---|---|---|---|---|---|---|
| Treatment effect | 0.0162 | 0.0193 | 0.00320 | 0.164*** | 0.0189 | 0.0218 | 0.050*** |
| (unadjusted) | (0.0147) | (0.0137) | (0.00667) | (0.0449) | (0.0143) | (0.0138) | (0.0124) |
| Adjusted treatment effect | 0.0157 | 0.0198 | 0.00384 | 0.140*** | 0.0162 | 0.0192 | 0.0468*** |
|  | (0.0145) | (0.0134) | (0.00638) | (0.0410) | (0.0138) | (0.0133) | (0.0118) |
| Control group average | 0.455 | 0.396 | 0.0718 | 2.956 | 0.474 | 0.396 | 0.396 |
| Survey modality | In person | In person | In person | Phone | Phone | Phone | Phone |
| Observations | 4,877 | 4,877 | 4,877 | 5,233 | 5,233 | 5,233 | 5,233 |

*Notes:* Coefficients displayed are mean differences with standard errors in parentheses. Adjusted models control for age group (age<20, age 20–34 (ref), age>=35), educational attainment, district, baseline family planning knowledge, baseline hemoglobin level and baseline body mass index. Numbers in parentheses are cluster-robust standard errors. Asterisks indicate p-value range of estimated differences: *p-value<0.10, **p-value<0.05, ***p-value<0.01. a) The full list of contraceptives is shown in Supplemental Materials S2 File. Contraceptives not considered as "modern" are beads/calendar method, withdrawal and breastfeeding.

Table 9 shows estimated impact on biomarkers. While we see some lower rates of anemia among treated women who currently are, or recently were pregnant (Table 9, column 2), these differences are not statistically significant. No changes were found for BMI overall, or any of the categories analyzed (underweight, overweight, obese).

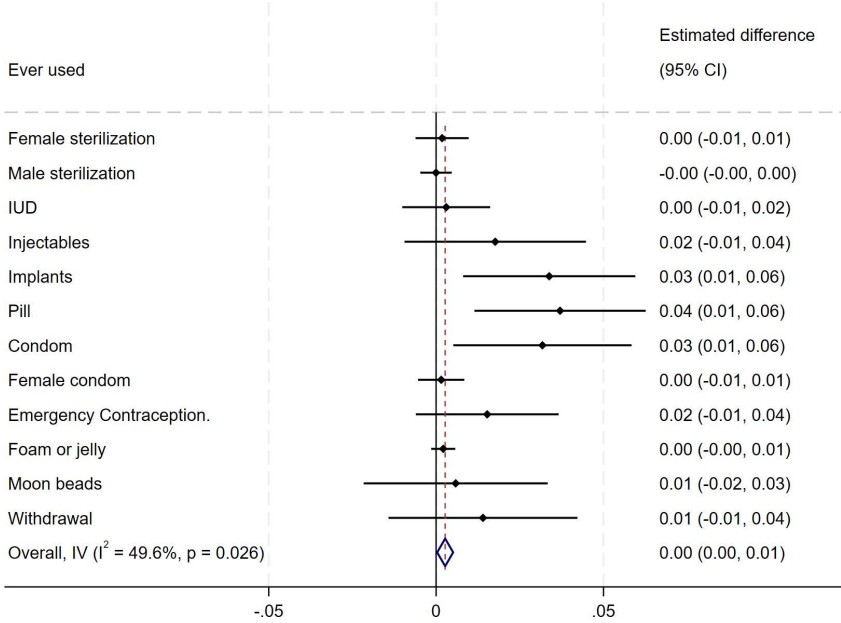

**Fig 4. Estimated impact on contraceptives ever used.** Figure notes: each point/square on the graphic represents the estimated coefficient obtained from regressing a binary indicator of having ever used the method in question on treatment – lines illustrate 95% confidence intervals.

**Table 6. Impact on prenatal care.**

| Outcome | Attended antenatal care | Currently taking iron | Deworming | Antenatal care for recent pregnancy | Recent pregnancy >=4 visits | Recent pregnancy: iron | Recent pregnancy: deworming |
|---|---|---|---|---|---|---|---|
| Treatment effect | 0.0593 | 0.0845* | 0.0167 | 0.00955 | 0.0229 | 0.0361* | 0.0374 |
| (unadjusted) | (0.0449) | (0.0431) | (0.0459) | (0.0161) | (0.0414) | (0.0189) | (0.0314) |
| Adjusted treatment effect | 0.0453 | 0.0758* | 0.00757 | 0.0156 | 0.0209 | 0.040** | 0.0369 |
|  | (0.0468) | (0.0456) | (0.0483) | (0.0158) | (0.0412) | (0.0189) | (0.0310) |
| Control group average | 0.621 | 0.605 | 0.451 | 0.958 | 0.454 | 0.901 | 0.808 |
| Sample | Women currently pregnant | | | Women completing pregnancy last year | | | |
| Survey modality | In person | | | | | | |
| Observations | 398 | 398 | 398 | 653 | 619 | 653 | 647 |

*Notes:* Coefficients displayed are mean differences with standard errors in parentheses. Adjusted models control for age group (age < 20, age 20–34 (ref), age >=35), educational attainment, district, baseline family planning knowledge, baseline hemoglobin level and baseline body mass index. Numbers in parentheses are cluster-robust standard errors. Asterisks indicate p-value range of estimated differences: *p-value<0.10, **p-value<0.05, ***p-value<0.01

Table 10 shows the result of the per protocol analysis, where we compare control women who had never called the platform to treated women who had called the platform at least five times. As expected, estimated impacts on knowledge scores are higher among compliers; however, these estimates become substantially smaller again when we adjust for women's differences in education and reproductive health knowledge at baseline.

## Stratified Results

Table 11 shows stratified impact results by age and educational attainment group, as well as by district. For age, we find the weakest effects among women under age 20, and largest impacts among women 35 years and older. Educational

**Table 7. Impact on Other Health Behaviors.**

| Outcome | Aims for health diet | Tried to lose weight | Tried to gain weight | Times at fruit | Times ate vegetables | Times ate protein | Currently taking iron supplements | Exercised |
|---|---|---|---|---|---|---|---|---|
| Treated | 0.0407*** | −0.025** | 0.0355*** | 0.0686** | 0.0484** | −0.00060 | 0.0156 | 0.0154 |
| (Unadjusted) | (0.0144) | (0.0108) | (0.0128) | (0.0346) | (0.0245) | (0.0183) | (0.00988) | (0.0124) |
| Adjusted treatment effect | 0.0312** | −0.025** | 0.0316** | 0.0583* | 0.0371* | 0.00202 | 0.0174* | 0.0119 |
| | (0.0123) | (0.0106) | (0.0126) | (0.0298) | (0.0213) | (0.0172) | (0.00965) | (0.0118) |
| Control group average | 0.552 | 0.255 | 0.242 | 1.335 | 1.332 | 1.760 | 0.136 | 0.335 |
| Survey modality | Phone survey | | | | | | | |
| Observations | 5,233 | 5,233 | 5,233 | 5,233 | 5,232 | 5,233 | 5,233 | 5,233 |

Notes: Coefficients displayed are mean differences with standard errors in parentheses. Adjusted models control for age group (age<20, age 20–34 (ref), age>=35), educational attainment, district, baseline family planning knowledge, baseline hemoglobin level and baseline body mass index. Numbers in parentheses are cluster-robust standard errors. Asterisks indicate p-value range of estimated differences: *p-value<0.10, **p-value<0.05, ***p-value<0.01

**Table 8. Impact on pregnancy and birth outcomes.**

| Outcome | Pregnancy was unplanned | Birth spacing<2 years | Previous pregnancy unplanned | Previous pregnancy low birthweight | Previous pregnancy lost |
|---|---|---|---|---|---|
| Treatment effect | −0.0605 | 0.0519 | 0.0260 | −0.0143 | −0.0301 |
| (unadjusted) | (0.0481) | (0.0546) | (0.0341) | (0.0246) | (0.0216) |
| Adjusted treatment effect | −0.0531 | 0.0496 | 0.0215 | −0.0157 | −0.0305 |
| | (0.0495) | (0.0516) | (0.0337) | (0.0245) | (0.0209) |
| Control group average | 0.410 | 0.372 | 0.261 | 0.0888 | 0.0384 |
| Survey modality | In person | | | | |
| Sample | Currently pregnant | | Completed pregnancy recently | | |
| Observations | 398 | 371 | 653 | 565 | 653 |

Notes: Coefficients displayed are mean differences with standard errors in parentheses. Adjusted models control for age group (age<20, age 20–34 (ref), age>=35), educational attainment, district, baseline family planning knowledge, baseline hemoglobin level and baseline body mass index. Numbers in parentheses are cluster-robust standard errors. Asterisks indicate p-value range of estimated differences: *p-value<0.10, **p-value<0.05, ***p-value<0.01.

**Table 9. Impact on biomarkers.**

| Outcome | Anemic (Hg<12 g/dL) | Anemic (Hg<12 g/dL) | Body mass index | Overweight (BMI>=25,<30) | Obese (BMI>=30) | Underweight (BMI<18.5) |
|---|---|---|---|---|---|---|
| Treatment effect | 0.00745 | −0.0495 | 0.183 | 0.00277 | 0.0150 | 0.0120 |
| (unadjusted) | (0.0129) | (0.0315) | (0.165) | (0.0126) | (0.0106) | (0.00826) |
| Adjusted treatment effect | 0.0111 | −0.0468 | 0.0922 | 0.000613 | 0.0102 | 0.0140* |
| | (0.0127) | (0.0311) | (0.111) | (0.0121) | (0.00831) | (0.00808) |
| Control group average | 0.23 | 0.23 | 24.89 | 0.26 | 0.15 | 0.08 |
| Sample | All | Current or recent pregnancy | All | All | All | All |
| Observations | 4,664 | 992 | 4,852 | 4,852 | 4,852 | 4,852 |

Notes: Coefficients displayed are mean differences with standard errors in parentheses. Adjusted models control for age group (age<20, age 20–34 (ref), age>=35), educational attainment, district, baseline family planning knowledge, baseline hemoglobin level and baseline body mass index. Numbers in parentheses are cluster-robust standard errors. Asterisks indicate p-value range of estimated differences: *p-value<0.10, **p-value<0.05, ***p-value<0.01.

**Table 10. Per protocol analysis.**

| Outcome | SRH Knowledge z-score | Currently uses modern method | Uses iron supplements | Used iron supplements | Lost pregnancy | Anemic | Anemic post pregnancy |
|---|---|---|---|---|---|---|---|
| Treatment effect | 0.127*** | 0.0409** | 0.0571 | 0.000959 | −0.00654 | 0.00586 | −0.0496 |
| (Unadjusted) | (0.0332) | (0.0159) | (0.0515) | (0.0102) | (0.00671) | (0.0147) | (0.0345) |
| Adjusted treatment | 0.0650** | 0.0206 | 0.0575 | 0.0117 | −0.00911 | 0.0168 | 0.0168 |
| effect | (0.0308) | (0.0155) | (0.0523) | (0.00954) | (0.00668) | (0.0144) | (0.0144) |
| Control group average | −0.0613 | 0.374 | 0.643 | 0.915 | 0.0386 | 0.229 | 0.389 |
| Sample | All | All | Pregnant now | Pregnant last year | | All | Pregnant |
| Restriction | Compliers only | | | | | | |
| Observations | 3,561 | 3,561 | 292 | 3,313 | 3,313 | 3,400 | 727 |

Notes:*Only women in the control group who never called the service, and women in the treatment group who called the service at least five times are included in this table. Coefficients displayed are mean differences with standard errors in parentheses. Adjusted models control for age group (age<20, age 20–34 (ref), age>=35), educational attainment, district, baseline family planning knowledge, baseline hemoglobin level and baseline body mass index. Numbers in parentheses are cluster-robust standard errors. Asterisks indicate p-value range of estimated differences: *p-value<0.10, **p-value<0.05, ***p-value<0.01. SRH = sexual and reproductive health.

**Table 11. Stratified results.**

| Stratum | Age<20 | Age 20–34 | Age 35+ | Secondary or higher education | Primary education | Kampala | Katakwi | Madi-Okollo | Rwampara |
|---|---|---|---|---|---|---|---|---|---|
| Treatment effect | 0.0213 | 0.0674* | 0.0901* | 0.0723* | 0.0587* | 0.0949** | 0.0828* | 0.122* | 0.0221 |
| | (0.127) | (0.0344) | (0.0500) | (0.0430) | (0.0331) | (0.0431) | (0.0455) | (0.0654) | (0.0517) |
| Observations | 291 | 2,978 | 1,608 | 1,804 | 3,073 | 1,472 | 1,093 | 650 | 1,662 |

*Notes:* Coefficients displayed are mean differences with standard errors in parentheses. Adjusted models control for age group (age<20, age 20–34 (ref), age>=35), educational attainment, district, baseline family planning knowledge, baseline hemoglobin level and baseline body mass index. Numbers in parentheses are cluster-robust standard errors. Asterisks indicate p-value range of estimated differences: *p-value<0.10, **p-value<0.05, ***p-value<0.01.

level seemed to have little impact, with only slightly larger effects for women with secondary or higher education compared to women with less education. At the district level, impacts vary substantially, with the largest treatment effects in Madi-Okollo (the 0.12 SD increase here is almost twice as large as the effect we see in the sample overall), and the smallest impacts in Rwampara (0.02 SD).

## Harms statement

No harm was reported during the trial.

## Discussion

The main objective of this study was to assess the impact of Viamo's call-in platform on health knowledge, behaviors and outcomes. Conceptually, knowledge platforms could result in changes on three distinct levels: changes in health knowledge, changes in health behavior, and changes in health outcomes. The results presented here suggest that changes in knowledge definitely occurred, with treated women displaying improved sex and reproductive health knowledge, being better informed about anemia treatment and more familiar with a larger range of contraceptive methods. This increased familiarity with contraceptive knowledge was also partially visible in contraceptive use. Treated women were on average only marginally (and insignificantly) more likely to use modern contraception, but used a different mix of contraceptives,

with higher usage of implants, and somewhat reduced reliance on injectables and calendar methods. In terms of future plans, the intervention seems to have substantially increased the proportion of women who say they will use modern contraception in the future.

For anemia – one of the key health challenges addressed on the platform – the treatment seems to have resulted in an increased uptake of supplements during pregnancy; these higher rates of supplementation also seem consistent with marginally lower rates of anemia among recently pregnant women. The estimated differences in anemia in this subgroup are however not statistically significant, which may partially be due to the relatively small number of women. Despite the high rates of anemia among non-pregnant women, we did not see any change in general protein intake or consumption of iron supplements. Given this lacking change (and the small proportion of women benefiting from additional supplementation during pregnancy), the insignificant impact on general hemoglobin levels makes sense.

We also see some interesting declines in the rates of pregnancy loss and the prevalence of low-birth-weight births in this subsample, but did not have sufficient statistical power here due to the relatively small number of women completing a pregnancy within the study period.

Given that a larger proportion of women in the treatment group claimed to aim for a healthy diet and to eat more fruit and vegetables, nutrition related content clearly reached its target. Interestingly, the messages received around "healthy body shapes" do not seem to have triggered a major shift towards eating less and lower BMIs, but seems to have induced a desire to change in both directions, i.e., we see an increasing proportion of women who want to lose weight and an increasing proportion of women who want to gain weight (and hence a smaller proportion of women who want to stay as they are). These shifts in intentions and behaviors are not seen in the anthropometric data – the intervention neither changed average BMI nor the proportions of women classified as underweight, overweight or obese.

In terms of the primary outcome, our results seem consistent with a recent review of mHealth intervention, which suggests that digital information sources can improve sexual and reproductive health knowledge as well as use of sexual and reproductive health services [16]. Compared to the estimates reported for an interactive intervention in Ghana [11,17] the effect sizes observed here seem smaller, which seems plausible given that not all women engaged with the service on a regular basis. Compared to previous studies conducted in other parts of Africa we also observed rather high sexual and reproductive health knowledge at baseline – most women were for example familiar with a range of contraceptive methods (beyond traditional methods, condoms and pills), and myths regarding side effects of contraception were less common that seen in other settings [11,17] – this likely also contributes to the somewhat smaller knowledge impacts seen here.

In terms of policy, one of the main advantages of the platform used is its relatively low cost – particularly when run at large scale; most of the costs of content development are related to the initial set-up, while operating costs are relatively minor. More research is needed here to investigate the cost-effectiveness of these new mobile technology channels as tools to improve health knowledge among vulnerable populations relative to traditional channels such as radio, tv or community outreach activities.

Despite the very large, stratified and regionally representative sample, the study has some limitations. First, there were not as many pregnant women as anticipated, reducing power on pregnancy-related behaviors and outcomes to some extent. Second, we found that the proportion of women using Airtel phones was substantially lower in some rural districts, making it very difficult to reach enrollment targets in some communities. We compensated this by enrolling more women in clusters with higher coverage. Even though a more selected sample clearly does not undermine the internal validity of the study (since randomization was done at the individual level), it is not clear whether the final set of women enrolled in the study is fully representative of the larger populations in the four study districts; it is also not clear if the four districts are fully representative of the country as a whole. Since the service is currently only available to Airtel users, this study simply focused on women with access to this service. Even though it seems plausible that similar effects could be found with other phone providers, it is not clear if the results of this study can be generalized to the country as a whole or to

other countries. Third, many of the behaviors measured were self-reported; while we implemented verification measures by taking pictures of antenatal and child health cards and asking women for evidence of contraceptive packaging, such direct evidence was generally only available for a small minority of women, and could thus not be used in the analysis. It is clearly possible that access to the service changes norms and reporting bias, which could explain some of the behavioral changes reported and detected in our analysis. The primary outcome of the study – the reproductive knowledge score – should not be subject to any reporting bias as all women should have the incentive to give the correct answer on the true/false knowledge quiz. Fourth, the individual-level randomization made the study vulnerable to spillover effects. Qualitative feedback collected at the end of the study suggests that treated women both invited other community members – spouses, friends and other family members – to use the service and actively shared new information with them. Given that some of this shared information likely reached the control group, it seems likely that the impact reported here underestimate the true causal effect of the program. We controlled for direct service use by excluding active users in the per protocol analysis (Table 10) – this does however not capture direct sharing of knowledge among community members.

## Conclusions

Overall, the results presented here suggest that the Viamo platform can effectively improve health knowledge and to some extent also health behaviors among vulnerable populations. These changes in knowledge, awareness and intentions could plausibly also induce changes in health outcomes in the long run; longer-term studies will likely be needed to accurately measure such effects. Future research should also aim to quantify knowledge spillovers within communities as well as the cost-effectiveness of call-in platform like the one analyzed here relative to other communication channels such as TV, radio, posters or community-based activities.

## Supporting information

**S1 File. Inclusivity-in-global-research-questionnaire 10 Sep 2025.docx: inclusivity statement.**
(DOCX)

**S2 File. Supplemental Materials 11 Sep 2025.docx: additional figures and tables references in the main manuscript.**
(DOCX)

**S3 File. Replication package csv.tar: complete anonymized data files as well as Stata DO file producing all figures and tables.**
(TAR)

## Acknowledgments

We would like to thank all involved staff members at IPA Uganda as well as in the Viamo team for their efforts and continued support of the project. We are also grateful to the more than 6000 study participants for their willingness to participate in the study. Lastly, we would like to thank the USAID Development Innovation Ventures (DIV) team for their input and feedback throughout the project.

## Author contributions

**Conceptualization:** Günther Fink.

**Data curation:** Günther Fink.

**Formal analysis:** Günther Fink.

**Investigation:** Günther Fink.

**Methodology:** Günther Fink.

**Project administration:** Günther Fink, Jan Will, Ilyena Kozain.

**Supervision:** Jan Will.

**Visualization:** Günther Fink, Branwen Nia Owen, Ilyena Kozain.

**Writing – original draft:** Günther Fink, Branwen Nia Owen, Jan Will, Ilyena Kozain.

**Writing – review & editing:** Günther Fink, Branwen Nia Owen, Ilyena Kozain.

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
