## [Decision Letter · Decision Letter 0]

25 Jul 2025

Dear Dr. Fink,

Thank you for submitting your manuscript to *PLOS ONE* . Your manuscript was reviewed by three reviewers, and after careful consideration of their comments, we believe that the study has merit but does not fully meet *PLOS ONE*

We look forward to receiving your revised manuscript.

Kind regards,

Poshan Thapa, PhD

Academic Editor

PLOS ONE

Journal Requirements:

2. Please describe in your methods section how capacity to provide consent was determined for the participants in this study. Please also state whether your ethics committee or IRB approved this consent procedure. If you did not assess capacity to consent please briefly outline why this was not necessary in this case.

3. Please include a complete copy of PLOS’ questionnaire on inclusivity in global research in your revised manuscript. Our policy for research in this area aims to improve transparency in the reporting of research performed outside of researchers’ own country or community. The policy applies to researchers who have travelled to a different country to conduct research, research with Indigenous populations or their lands, and research on cultural artefacts. The questionnaire can also be requested at the journal’s discretion for any other submissions, even if these conditions are not met.  Please find more information on the policy and a link to download a blank copy of the questionnaire here: https://journals.plos.org/plosone/s/best-practices-in-research-reporting. Please upload a completed version of your questionnaire as Supporting Information when you resubmit your manuscript.

“This project was funded through USAID DIV grant number 7200AA21FA00018”

“The first, second and third author declare no competing interest. The last author (IK) is currently employed by Viamo, the company operating the service analyzed. She was not involved in the data analysis and did not modify the way the results are presented or described in any way.”

7. We note that Figure 1 in your submission contain [map/satellite] images which may be copyrighted. All PLOS content is published under the Creative Commons Attribution License (CC BY 4.0), which means that the manuscript, images, and Supporting Information files will be freely available online, and any third party is permitted to access, download, copy, distribute, and use these materials in any way, even commercially, with proper attribution. For these reasons, we cannot publish previously copyrighted maps or satellite images created using proprietary data, such as Google software (Google Maps, Street View, and Earth). For more information, see our copyright guidelines: http://journals.plos.org/plosone/s/licenses-and-copyright.

Reviewers' comments:

Reviewer's Responses to Questions

**Comments to the Author**

1. Is the manuscript technically sound, and do the data support the conclusions?

Reviewer #1: Partly

Reviewer #2: Partly

Reviewer #3: Partly

2. Has the statistical analysis been performed appropriately and rigorously?

Reviewer #1: I Don't Know

Reviewer #2: Yes

Reviewer #3: I Don't Know

3. Have the authors made all data underlying the findings in their manuscript fully available?

Reviewer #1: Yes

Reviewer #2: No

Reviewer #3: Yes

4. Is the manuscript presented in an intelligible fashion and written in standard English?

Reviewer #1: No

Reviewer #2: Yes

Reviewer #3: Yes

Reviewer #1: Thank you for sharing this very interesting study for review. Below are some points that might help the authors improve their manuscript:

Introduction:

- It would be beneficial to mention a clear hypothesis or research question at the end of the introduction. This will guide readers in understanding how the results align with the study’s goals.

Methods:

- The methodology is well-designed and aligns with the CONSORT checklist. However, further details are needed regarding the data collection tools. Specifically:

- Which tools were used to assess the primary and secondary variables?

- Validity and reliability of the measurement tools should be addressed to ensure the robustness of the results.

- The term “normalized” is used to describe the data collection tool. This terminology is not common in this context. Consider replacing it with terms like “valid and reliable.”

- The terms *“reproductive health knowledge” and “antenatal care”seem to overlap or create confusion. Please clarify the distinction between these terms and ensure consistency throughout the manuscript.

- Details regarding exclusion criteria are missing and should be provided.

- Clarify who assessed eligibility and whether they were blinded during the randomization process.

- The manuscript contains 11 tables, which may be excessive. Consider merging some tables or limiting the number to those most essential for clarity. Ensure all tables mentioned in the manuscript are explained in the results section.

- There is no mention of the interventional materials used with the experimental group. This information is necessary for understanding the study design and replication in future research.

- The discussion section lacks comparisons with previous studies. Including these would contextualize the findings and add depth.

- Discuss the study’s limitations and strengths to aid future researchers.

- The trial flowchart should be reshaped to align with the original CONSORT 2010 format. available at CONSORT 2010

Reviewer #2: I commend the authors' work to assess the impact of Viamo’s call-in platform on sexual and reproductive health

knowledge, behaviors and outcomes.

Please find my comments below (pg numbers used in the absence of line numbers):

-pg 1/27: Title: Please consider revising the title to reflect the aim of this manuscript and the analysis/discussion/conclusion. For instance, "Estimating the Impact of Viamo Call-in Information Services on Sexual and Reproductive Health Knowledge, Behavior, and Outcomes among women of reproductive age: Results from a 2-Arm Open Label Randomized Controlled Trial in Uganda" may be a more appropriate title

-pg 1/27: Please maintain a consistent spelling for "behavior". In some aspects of the manuscript, this is spelt at "behaviour"

-pg 2/27: Please cite this source of this sentence, "The platform is free of charge, and currently has over 4 million users in Uganda, and 42 million users in LMICs globally."

-pg 9/27: "Follow-up rates (completing either the phone or the in person survey or both) were 92% in both arms." Please review this sentence and consider removing it since the estimates do not correspond with Figure 2 and the other sentences in this section of the manuscript better interpret the numbers.

-pg 11/27: Table 2: Please review the results with *** (p-value <0.01). All the P-values are >0.01 and ** (p-value<0.05) would be more appropriate. In addition, some p-values are not significant (0.471 and 0.958) and should not have any asterix on them.

-pg 12/27: Table 3 footnote: How was C2L calculated? It doesn’t seem to Times listened / Times reached based on the table. Please add the formula to the footnote.

-pg 13/27: Table 4: Please review the result with *** (p-value <0.01) because ** (p-value<0.05) would be more appropriate for p-value 0.0265 – row 2, column 2.

-pg 15/27: Table 5: Please review the results with *** (p-value <0.01). All the P-values are >0.01 but <0.05 and ** (p-value<0.05) would be more appropriate.

-pg 16/27: Table 6: Please review the results with * (p-value <0.01). All of them are <0.05 and ** (p-value<0.05) would be more appropriate.

-pg 17/27: Table 7: Please review the results with *** (p-value <0.01) and * (p-value <0.10) and correct them as appropriate. All the p-values should have ** except p-value for 0.0174 (0.00965) which should have ***

-pg 18/27: Table 8: Please include the corresponding asterix for the estimates with significant p-values in this table.

-pg 19/27: Table 9: Please include the corresponding asterix for the estimates with significant p-values in this table.

-pg 20/27: Table 10: Please include the corresponding asterix for the estimates with significant p-values in this table.

-pg 21/27: Table 11: Please review the results with p-values 0.0344*, 0.0331* and 0.0455* (where * is p-value <0.10) and correct them to ** (p-value <0.05) as appropriate. All should have ** except p-value: 0.00965 which should have ***

-pgs 22-24: Discussion: Please incorporate existing studies that support or dispute your findings in the discussion section. This section needs a lot more work and is currently only a repetition/elaboration of the results' interpretation.

Reviewer #3: I struggled to review this paper specifically the results section.

I could not figure out what metric was used to measure impact (such odd ratios).

The measurement of the outcome is not clear or easily understood.

**Do you want your identity to be public for this peer review?** For information about this choice, including consent withdrawal, please see our Privacy Policy

Reviewer #1: **Yes:**  Fatemeh Zarei

Reviewer #2: No

Reviewer #3: No

---

## [Author Response · Author response to Decision Letter 1]

21 Oct 2025

and

We have reviewed the manuscript and reformatted the text following the PLOS one style templates.

2. Please describe in your methods section how capacity to provide consent was determined for the participants in this study. Please also state whether your ethics committee or IRB approved this consent procedure. If you did not assess capacity to consent please briefly outline why this was not necessary in this case.

We read out the consent form to illiterate women and asked them to provide a thumb print to express their explicit consent instead of signing. This process was suggested and approved by local IRB. Women not able to understand the scope of the study as well as women not able or willing to consent using either modality were excluded from the study.

3. Please include a complete copy of PLOS’ questionnaire on inclusivity in global research in your revised manuscript. Our policy for research in this area aims to improve transparency in the reporting of research performed outside of researchers’ own country or community. The policy applies to researchers who have travelled to a different country to conduct research, research with Indigenous populations or their lands, and research on cultural artefacts. The questionnaire can also be requested at the journal’s discretion for any other submissions, even if these conditions are not met. Please find more information on the policy and a link to download a blank copy of the questionnaire here: https://journals.plos.org/plosone/s/best-practices-in-research-reporting. Please upload a completed version of your questionnaire as Supporting Information when you resubmit your manuscript.

We have completed the inclusivity questionnaire and have uploaded it with the revised documents.

“This project was funded through USAID DIV grant number 7200AA21FA00018”

This is correct - we have included this statement in the cover letter as suggested.

“The first, second and third author declare no competing interest. The last author (IK) is currently employed by Viamo, the company operating the service analyzed. She was not involved in the data analysis and did not modify the way the results are presented or described in any way.”

We have included an updated conflict of interest statement in the cover letter.

We have uploaded a complete replication package comprising both the data files and the Stata DO files creating figures and tables as supplementary materials.

7. We note that Figure 1 in your submission contain [map/satellite] images which may be copyrighted. All PLOS content is published under the Creative Commons Attribution License (CC BY 4.0), which means that the manuscript, images, and Supporting Information files will be freely available online, and any third party is permitted to access, download, copy, distribute, and use these materials in any way, even commercially, with proper attribution. For these reasons, we cannot publish previously copyrighted maps or satellite images created using proprietary data, such as Google software (Google Maps, Street View, and Earth). For more information, see our copyright guidelines: http://journals.plos.org/plosone/s/licenses-and-copyright.

This shapefile comes from GADM, which explicitly allows such use: “Using the data to create maps for publishing of academic research articles is allowed. Thus you can use the maps you made with GADM data for figures in articles published by PLoS, Springer Nature, Elsevier, MDPI, etc. You are allowed (but not required) to publish these articles (and the maps they contain) under an open license such as CC-BY as is the case with PLoS journals and may be the case with other open access articles.” (https://gadm.org/license.html),

We have added the source to the legend of Figure 1.

We have relabeled the supplemental material figures following these guidelines.

NA – we did not received any such requests.

Reviewers' comments:

Reviewer #1:

Thank you for sharing this very interesting study for review. Below are some points that might help the authors improve their manuscript.

Thank you for the careful review and very helpful suggestions – we have done our best to address all of them as outlined in further detail below.

Introduction:

- It would be beneficial to mention a clear hypothesis or research question at the end of the introduction. This will guide readers in understanding how the results align with the study’s goals.

Thank you for this suggestion. We have added the following sentence to the Introduction:

“Our primary hypothesis was that access to the platform can improve sexual and reproductive health knowledge and attitudes.”

Methods:

- The methodology is well-designed and aligns with the CONSORT checklist. However, further details are needed regarding the data collection tools. Specifically:

- Which tools were used to assess the primary and secondary variables?

Apologies for the lack of detail here in the original manuscript. We provide further details on the outcomes variables on page 8 and 9 now, where we write:

“The two primary outcomes of the study were reproductive health knowledge and nutritional supplementation in pregnancy. For reproductive health, we used a 15-item questionnaire of true/false statements (items listed in Supplemental Materials S2). This tool was based on previous studies conducted in sub-Saharan Africa [7] and adapted to match local context. To facilitate interpretation of regression results, we converted the raw scores (number of correct answers out of 15) into z-scores with mean zero and standard deviation one. For antenatal care, we used self-reported iron supplementation as an outcome measure. Secondary outcomes included contraceptive use, prenatal care uptake, hemoglobin level (as a biomarker indicating actual iron intake), body mass index, and birth spacing.

For pregnancy related outcomes, we analyzed two separate sets of women: first, women reporting to be pregnant at the time of the endline survey (currently pregnant), and second, women who completed a pregnancy in the year preceding the endline survey (recently pregnant). For current pregnancies, we analyzed self-report iron supplementation as well as self-reported use of antenatal care services, and whether women indicate the pregnancy was planned. For recently completed pregnancies, we analysed the same variables, but also analysed pregnancy loss and low birth weight (reported weight at birth < 2.5kgs) as additional secondary outcomes.

- Validity and reliability of the measurement tools should be addressed to ensure the robustness of the results.

We added some more details in the Methods section, where we write:

“This tool was based on previous studies conducted in sub-Saharan Africa [7] and adapted to match local context.”

For the secondary outcomes, self-reporting biases are definitely a concern which we discuss in the Limitations section of the paper, where we write:

“Third, many of the behaviors measured were self-reported; while we implemented verification measures by taking pictures of antenatal and child health cards and asking women for evidence of contraceptive packaging, such direct evidence was generally only available for a small minority of women, and could thus not be used in the analysis. It is clearly possible that access to the service changes norms and reporting bias, which could explain some of the behavioral changes reported and detected in our analysis. The primary outcome of the study - the reproductive knowledge score - should not be subject to any reporting bias as all women should have the incentive to give the correct answer on the true/false knowledge quiz.”

- The term “normalized” is used to describe the data collection tool. This terminology is not common in this context. Consider replacing it with terms like “valid and reliable.”

Our apologies – we just meant to say that we created a z-score out of the raw number of correct answers prior to analysis. We have rephrased this as follows in the revised Methods section:

“To facilitate interpretation of regression results, we converted the raw scores (number of correct answers out of 15) into z-scores with mean zero and standard deviation one.”

- The terms “reproductive health knowledge” and “antenatal care” seem to overlap or create confusion. Please clarify the distinction between these terms and ensure consistency throughout the manuscript.

Apologies for the confusion: we meant to refer to nutritional supplementation during antenatal care here, and have made this more explicit in the revised version of the manuscript where we write:

“For nutritional supplementation during pregnancy, we used self-reported iron supplementation as an outcome measure”

- Details regarding exclusion criteria are missing and should be provided.

Apologies for the lack of detail here – we have added a new section on inclusion and exclusion criteria, which reads as follows:

“..Study eligibility was assessed by trained study enumerators. Women living in selected enumeration areas were eligible for inclusion in the study if they were aged 18-49 years and had access to an Airtel (a mobile network operator) phone number within their household. Given the explicit focus on antenatal care, family planning and post-partum, all women who were pregnant during the listing were automatically invited to join the study; non-pregnant women were randomly selected for the study until the 45 women per cluster target was reached. In clusters with less than 45 women, all women were invited to participate in the study. Women were ineligible if they had used the Viamo platform previously, they planned to move outside of the study area within 12 months of baseline or if they did not speak one of the six languages available on the platform: Ateso, English, Luganda, Lugbara, Luo or Runyakitara.”

- Clarify who assessed eligibility and whether they were blinded during the randomization process.

We have added the following information:

“Study eligibility was assessed by trained study enumerators.”

- The manuscript contains 11 tables, which may be excessive. Consider merging some tables or limiting the number to those most essential for clarity. Ensure all tables mentioned in the manuscript are explained in the results section.

We agree that there is lots of information in this manuscript, but really believe that all of these results are important. Given that there is no formal limit at PLOS One, we would thus prefer to leave all figures and tables in, and have verified all of them are mentioned in the text. If you feel strongly that some figures should go to the Supplemental Materials we would of course be happy to accommodate that.

- There is no mention of the interventional materials used with the experimental group. This information is necessary for understanding the study design and replication in future research.

We fully agree with this. We provide a detailed description of the intervention on page 6. The intervention is easy and complex at the same time: it is easy, because essentially women are just invited to call a toll-free number and then get to access content designed for them. The complexity comes from the fact that women can freely navigate content. We show the most commonly accessed items in Table 3, and provide a more detailed example in supplemental materials S1.

- The discussion section lacks comparisons with previous studies. Including these would contextualize the findings and add depth.

Thank you for this suggestion. We have added the following paragraph to the Disucssion:

“In terms of the primary outcome, our results seem consistent with a recent review of mHealth intervention, which suggests that digital information sources can improve sexual and reproductive health knowledge as well as use of sexual and reproductive health services [10]. Compared to the estimates reported for an interactive intervention in Ghana [7, 11] the effect sizes observed here seem smaller, which seems plausible given that not all women engaged with the service on a regular service. “

- Discuss the study’s limitations and strengths to aid future researchers.

We discuss strengths and limitatios at the very end of the Discussion section, where we write:

“Despite the very large, stratified and regionally representative sample, the study has some limitations. First, there were not as many pregnant women as anticipated, reducing power on pregnancy-related behaviors and outcomes to some extent. Second, we found that the proportion of women using Airtel phones was substantially lower in some rural districts, making it very difficult to reach enrollment targets in some communities. Even though a more selected sample clearly does not undermine the internal validity of the study (since randomization was done at the individual level), it is not clear whether the final set of women enrolled in the study is fully representative of the larger populations in the four study districts as well as Uganda overall - since the service is currently only available to

---

## [Decision Letter · Decision Letter 1]

18 Dec 2025

Dear Dr. Fink,

Thank you for submitting your manuscript to PLOS ONE. After careful consideration, we feel that it has merit but does not fully meet PLOS ONE’s publication criteria as it currently stands. Therefore, we invite you to submit a revised version of the manuscript that addresses the points raised during the review process.

We look forward to receiving your revised manuscript.

Kind regards,

Alfredo Luis Fort, M.D., M.Sc., Ph.D.

Academic Editor

PLOS One

Journal Requirements:

Additional Editor Comments (if provided):

You have made several additions and changes to the original manuscript. However, the revised version still needs a few clarifications and changes to make it more understandable and avoid confusion to the readers. Please see the suggestions in the attached files and improve the article once more to make it publishable. Thank you.

Reviewers' comments:

Reviewer's Responses to Questions

**Comments to the Author**

Reviewer #4: (No Response)

Reviewer #5: (No Response)

Reviewer #6: (No Response)

2. Is the manuscript technically sound, and do the data support the conclusions?

Reviewer #4: Partly

Reviewer #5: (No Response)

Reviewer #6: Yes

3. Has the statistical analysis been performed appropriately and rigorously?

Reviewer #4: No

Reviewer #5: (No Response)

Reviewer #6: Yes

4. Have the authors made all data underlying the findings in their manuscript fully available?

Reviewer #4: Yes

Reviewer #5: (No Response)

Reviewer #6: Yes

5. Is the manuscript presented in an intelligible fashion and written in standard English?

Reviewer #4: Yes

Reviewer #5: Yes

Reviewer #6: Yes

Reviewer #4: REVIEWER COMMENTS

Thank you for sharing this study for review. Below are some points that might help the authors revamp their manuscript.

Abstract

It will be beneficial to add dad analysis section to the methods section of the abstract. Which statistical test were used.

If you decide to retain vulnerable population in the title, it will be important to clarify which specific vulnerable population you studied. Otherwise, I will suggest that you remove the vulnerable population in the title and put “women of reproductive age”, and hence the background section will need to be revised.

I am curious about the choice of test used in the data analysis and why the continuous variables

Introduction

The first paragraph in this section sounds like a problem statement and also this entire section needs to be revised. I suggest that this section be revised as follows:

Introduce the reproductive age group as the predominant age group in Uganda

Health outcomes common among the reproductive age group (vulnerable population)

Emerging role of mobile phone app (Viamo Call-in Information Services)/internet coverage/technology in health education and improvement of reproductive and sexual health outcome in the Ugandan population. The proportion of the Ugandans with access to cell phones and/or internet. Previous studies in the impact of other cell phone apps on reproductive and sexual health outcome.

This section should now end with the problem statement and the rational for this study.

Methods

Study design: Were the 4 purposively selected districts representative of the entire Uganda population. This may act as a threat to external validity given that the 4 district doesn’t represent the minority regions. I guess this is a limitation.

Sampling technique: I think the sampling technique you described was a multistage cluster sampling technique and not a 2-stage random sampling. Also take a good look at different studies and revise the entire sampling procedure. The selection of 45 per cluster makes me worry if the selection was proportionate or representative each cluster? In order words, were the clusters of equal population for 45 to be selected from each clusters?

Sample size: How was the sample size of 6000 arrived at? It will make sense to add how it was calculated

Randomization: The section on randomization is very scanty and needs clarification. At this point its still not clear what the non-treatment arm received. Did they also benefit from Viamo Call-in Information Services, its important to clearly state what the intervention the 2 groups received. Its important to clearly define the intervention and the non-intervention group (treatment vs control group) and exactly what they received because its not clear.

Primary and secondary outcomes: The definition of primary and secondary outcomes (how were they measured) needs to be included in the methods section. Which tool was used to measure knowledge? Was it pretested or was it a validated tool?

Data analysis: I have noticed that the majority of the variables were analyzed as continuous variables. Did you consider converting the continuous variables (scores) to categorical variables which then permits you to use different methods.

Results

The presentation of results is not very clear. The tables are many and its not clear labelled if its correlation coefficient or a p-value.

Discussion

The discussion needs to be revised. It lack the comparison and contrast of the study findings with studies done locally and international.

Reviewer #5: I appreciate the submission of this manuscript. Please find my comments below, which may be helpful in improving the quality of your work:

Pg. 1/26: It is recommended that the phrase “results from” in the title be omitted, as this would shorten the title without detracting from its intended meaning.

Pg. 2/26:

- It is recommended that the phrase “and Outcomes” be included in the title, as currently presented at the end of the background, to ensure that the title is complete.

- Please ensure that the sampling method, data collection tools, a brief description of the data collection procedure, and the statistical tests used are clearly described in the methods section.

Pg. 3/26: Please mention keywords at the end of the abstract.

Pg. 4/26: Please provide further details regarding the Viamo platform and its operational mechanisms. Specifically, clarify whether the platform enables two-way interaction and whether users can experience a form of interactive support comparable to counseling, or whether information is delivered in a predominantly one-way manner, functioning mainly as a communication channel for educational purposes.

Please review relevant studies that have implemented similar interventions in the same research area, either locally or in other regions. Additionally, address studies that have evaluated the effectiveness of the Viamo platform or comparable mobile- or voice-based platforms for different purposes. Finally, summarize the existing literature and highlight the comparative advantage and necessity of the present intervention by identifying the gaps it aims to address. If no relevant studies are available, this should be clearly stated.

Pg. 5/26 (Methods section):

- Please specify the exact nature of any encouragement designs and whether they may have preferentially attracted certain subgroups of participants. For example, as noted on page 7, "Participants received UGX 10,000 (~USD $2.77) of phone credit as a token of gratitude after each survey interview." Offering such incentives could disproportionately attract individuals with lower socioeconomic status, potentially limiting the generalizability. Therefore, if participants were aware of the incentives at the time of study invitation, this should be explicitly stated and discussed in the study limitations.

- Please specify the exclusion (withdrawal) criteria during the study period, which criteria refer to circumstances that necessitate the removal of participants after enrollment, such as the death of a close family member.

- Please specify the sample size and describe the method used for its calculation in the Methods section.

Pg. 6/26:

- Please provide additional details on how participants used the Viamo platform. Clarify whether platform use was need-based and participant-driven or task-oriented as part of the intervention.

- You mentioned in the limitations that information leak or unintended use of the platform by control group participants may have occurred. Please clarify whether any measures were taken to limit access to the platform or prevent its use by the control group and, if so, describe these measures in the Methods section. In addition, indicate whether any promotional activities (such as social media or other communication channels) were used that might have increased awareness of the platform and led to its use among control group participants.

Pg. 7/26: Were there specific intervals between each measurement time point? If so, please specify these intervals.

Pg. 8/26:

- If the reproductive health questionnaire used in the study has a specific name or is a standardized tool recognized in the literature, please provide its official name and any relevant references.

- The current description of study variables appears unclear and potentially confusing. The title refers to Sexual and Reproductive Health Knowledge, Behavior, and Outcomes. The explanation provided under primary and secondary outcomes could clearly reflect the study objectives and variables if organized and presented systematically. For instance, the terms reproductive health "behavior" and "outcome" seem to overlap; if this is the case, consider removing one term from the title and text to maintain consistency. Otherwise, please provide a clear definition of each variable and explain how it is measured. Similarly, the term "sexual" may overlap with "reproductive health", as behaviors measured under "sexual behavior" might also fall under reproductive health outcomes. Therefore, please clarify this distinction or consider merging the two concepts where appropriate. A concise, organized, and precise description of variables and outcomes will improve understanding of the study’s objectives and interpretation of results.

Pg. 9/26: Please describe the descriptive and inferential statistical tests used in the study, and specify the statistical methods applied for the presentation and interpretation of the tables.

Pg. 11/26: Please mention p-values and place the asterisk indicators (*) next to the p-values only, and not next to other values. Consider this issue for all tables.

Pg. 13/26: In Table 4, you referred to “Beliefs that local herbs cure anemia”; however, this variable has not been mentioned in either the primary or secondary outcomes, nor in the measurement instruments described in the supplementary files. Please provide clarification on this issue.

Pg. 15/26: Since Survey modality is reported at the end of the table, please refrain from repeating it in the outcome rows.

Pg. 15-20: Was a specific checklist or questionnaire used to collect the data presented in Tables 5–10? Please clarify this point. If data collection and evaluation were based on a checklist, please describe it in the relevant sections of the manuscript.

Pg. 15-19: The outcomes reported in Tables 5, 7, 8, and 9 may have been influenced by pregnancy and postpartum healthcare recommendations, medication use, or being under the supervision of a physician or nutritionist. Please clarify how these potential confounding factors were addressed in the analyses. If any strategies were implemented to control for them, please describe them explicitly; otherwise, please acknowledge this issue in the limitations section. In addition, it is recommended that the results for pregnant and non-pregnant women be reported separately in these tables.

Pg. 22-24:

- Not all results have been fully interpreted, and the interpretation provided is rather brief. Please expand the Discussion section to provide a more comprehensive interpretation of the study findings. For instance, the observed reduction in pregnancy loss and low birth weight could be interpreted in relation to the level of antenatal care received. In addition, for each outcome discussed, please contextualize the findings within the existing literature by comparing them with results from similar studies and discussing areas of consistency or discrepancy.

- Please critically assess the generalization of the study findings and clarify the extent to which the results can be applied to other populations or settings.

Reviewer #6: (No Response)

**Do you want your identity to be public for this peer review?** For information about this choice, including consent withdrawal, please see our Privacy Policy

Reviewer #4: No

Reviewer #5: No

Reviewer #6: No

---

## [Author Response · Author response to Decision Letter 2]

19 Jan 2026

Reviewer #4:

Thank you for sharing this study for review. Below are some points that might help the authors revamp their manuscript.

Thank you for your kind and constructive comments – we have done our best to address all of them as outlined in detail below.

Abstract

It will be beneficial to add analysis section to the methods section of the abstract. Which statistical test were used.

Thank you for this suggestion. We have added the following sentence to the Abstract to clarify the empirical models used:

“Linear regression models were used to estimate mean differences in outcomes between treatment and control. Standard errors were corrected for the cluster-based sampling of women.”

If you decide to retain vulnerable population in the title, it will be important to clarify which specific vulnerable population you studied. Otherwise, I will suggest that you remove the vulnerable population in the title and put “women of reproductive age”, and hence the background section will need to be revised.

We have removed the “vulnerable populations” in the background section and now write:

“..knowledge remains limited among women of reproductive health in many low and middle income countries”

I am curious about the choice of test used in the data analysis and why the continuous variables

As stated above, we used linear regression models to estimate mean differences in outcomes between treated and controls. We also considered logistic regression models for the binary outcomes, but decided to stick with mean differences because they make comparisons in outcomes easy to communicate (and use cluster-robust standard errors to address non-normal residuals for binary outcomes).

Introduction

The first paragraph in this section sounds like a problem statement and also this entire section needs to be revised. I suggest that this section be revised as follows:

Introduce the reproductive age group as the predominant age group in Uganda

Health outcomes common among the reproductive age group (vulnerable population)

Emerging role of mobile phone app (Viamo Call-in Information Services)/internet coverage/technology in health education and improvement of reproductive and sexual health outcome in the Ugandan population. The proportion of the Ugandans with access to cell phones and/or internet. Previous studies in the impact of other cell phone apps on reproductive and sexual health outcome. This section should now end with the problem statement and the rational for this study.

We have restructured the Introduction as suggested and added several additional references. The first paragraph introduces the knowledge gap and highlights the health consequences of this gap for mothers and offsprings. The second paragraph summarizes current research on how to best improve knowledge among women of reproductive age. The third paragraph introduces cell phones as new tool and highlights key studies. The last one provides the objective and rationale of the study.

Methods

Study design: Were the 4 purposively selected districts representative of the entire Uganda population. This may act as a threat to external validity given that the 4 district doesn’t represent the minority regions. I guess this is a limitation.

We agree this is a limitation and acknowledge this in the revised Discussion section, where we write: .

“…it is not clear whether the final set of women enrolled in the study is fully representative of the larger populations in the four study; it is also not clear if the four districts are fully representative of the country as a whole.”

Sampling technique: I think the sampling technique you described was a multistage cluster sampling technique and not a 2-stage random sampling. Also take a good look at different studies and revise the entire sampling procedure. The selection of 45 per cluster makes me worry if the selection was proportionate or representative each cluster? In order words, were the clusters of equal population for 45 to be selected from each clusters?

We have changed the description to multi-stage sampling. The sampling was not proportional to population size and likely did not generate a fully representative sample – we say this clearly in the Discussion, where we write:

“Second, we found that the proportion of women using Airtel phones was substantially lower in some rural districts, making it very difficult to reach enrollment targets in some communities. We compensated this by enrolling more women in clusters with higher coverage. Even though a more selected sample clearly does not undermine the internal validity of the study (since randomization was done at the individual level), it is not clear whether the final set of women enrolled in the study is fully representative of the larger populations in the four study districts.”

Sample size: How was the sample size of 6000 arrived at? It will make sense to add how it was calculated

We explain this in the revised Power Calculations section, where we write:

“The study was powered to detect a 0.1 standard deviation (SD) difference in standardized knowledge scores (mean 0, SD 1) with power 0.9. Assuming an average cluster size of 30, a follow-up rate of 90% and a design effect of 1.3, this required 100 clusters per arm, and a total sample size of 6000 women.”

Randomization: The section on randomization is very scanty and needs clarification. At this point its still not clear what the non-treatment arm received. Did they also benefit from Viamo Call-in Information Services, its important to clearly state what the intervention the 2 groups received. Its important to clearly define the intervention and the non-intervention group (treatment vs control group) and exactly what they received because its not clear.

Apologies for the lack of clarity. We have added the following text to the Randomization section:

“Women selected for the treatment arm were then provided with a personalized introduction to the service (after the baseline survey was completed) and called the service directly from their own phone with the support of the interviewer to see how the system works. Treated women were also given a promotional calendar to serve as a physical reminder on how to navigate the service and the benefits of the service. Women in the control group were not given any intervention. Given that the service was freely available to all women, they were in theory able to call the service at any time, but not given any encouragement or support to do so by the study team.”

Primary and secondary outcomes: The definition of primary and secondary outcomes (how were they measured) needs to be included in the methods section. Which tool was used to measure knowledge? Was it pretested or was it a validated tool?

Apologies for the lack of detail here. We now explain this in more detail in the revised Methods section:

“The two primary outcomes of the study were reproductive health knowledge and nutritional supplementation in pregnancy. For reproductive health, we used a 15-item questionnaire of true/false statements (items listed in Supplemental Materials S2). This tool was based on previous studies conducted in sub-Saharan Africa [11] and pretested locally to make sure all items were suitable for the study population. To facilitate interpretation of regression results, we converted the raw scores (number of correct answers out of 15) into z-scores with mean zero and standard deviation one. For nutrition knowledge, we analyzed a binary indicator for women (falsely) believing that herbs can cure anemia. For nutritional supplementation during pregnancy, we used self-reported iron supplementation as an outcome measure. Secondary outcomes included contraceptive knowledge (number of methods respondents are familiar with), contraceptive use (currently using any modern method), prenatal care uptake (any, number of visits), hemoglobin level (as a biomarker indicating actual iron intake), body mass index, and birth spacing.

For pregnancy related outcomes, we analyzed two separate sets of women: first, women reporting to be pregnant at the time of the endline survey (currently pregnant), and second, women who completed a pregnancy in the year preceding the endline survey (recently pregnant). For current pregnancies, we analyzed self-report iron supplementation as well as self-reported use of antenatal care services, and whether women indicate the pregnancy was planned. For recently completed pregnancies, we analyzed the same variables, but also analyzed pregnancy loss and low birth weight (reported weight at birth < 2.5kgs) as additional secondary outcomes. “

Data analysis: I have noticed that the majority of the variables were analyzed as continuous variables. Did you consider converting the continuous variables (scores) to categorical variables which then permits you to use different methods.

We have explained this in further detail in the revised Methods section, where we write:

“All primary models were estimated intent-to-treat, comparing outcomes between treated and control women at endline. For all outcomes, we estimated both unadjusted linear regression models (not including any covariates) and fully adjusted models, where we controlled for age, district, marital status, educational attainment and baseline knowledge variables. The main rationale for using linear regression models was to generate mean differences estimates that could be compared across the rather large set of outcome variables. To adjust for non-normal residuals, we used Huber’s cluster-robust standard errors [15].”

Results

The presentation of results is not very clear. The tables are many and its not clear labelled if its correlation coefficient or a p-value.

We agree that there are many, but think it is important to show all of the results – also one of the reasons we chose PLOS ONE for this paper. We have added notes to all tables to be clear about what is shown (coefficients, SEs, p-values). For example, for the first regression Table (T2), the notes now say:

“All point estimates displayed correspond to estimated mean differences in outcomes. Numbers in parentheses are cluster-robust standard errors. Asterisks indicate p-value range of estimated differences: *p-value<0.10, **p-value<0.05, ***p-value<0.01”

Discussion

The discussion needs to be revised. It lack the comparison and contrast of the study findings with studies done locally and international.

There are not so many studies of similar nature, but we try to compare the main outcomes to the related literature in the Discussion, where we write:

“In terms of the primary outcome, our results seem consistent with a recent review of mHealth intervention, which suggests that digital information sources can improve sexual and reproductive health knowledge as well as use of sexual and reproductive health services [16]. Compared to the estimates reported for an interactive intervention in Ghana [11, 17] the effect sizes observed here seem smaller, which seems plausible given that not all women engaged with the service on a regular service. Compared to previous studies conducted in other parts of Africa we also observed rather high sexual and reproductive health knowledge at baseline – most women were for example familiar with a range of contraceptive methods (beyond traditional methods, condoms and pills), and myths regarding side effects of contraception were less common that seen in other settings [11, 17] – this likely also contributes to the somewhat smaller knowledge impacts seen here. “

Please let us know if there are any papers that we missed and should be cited here.

Reviewer #5:

I appreciate the submission of this manuscript. Please find my comments below, which may be helpful in improving the quality of your work.

Thank you for the kind and helpful comments – we really appreciate your input and have done our best to address all suggestions and explained in further detail below.

Pg. 1/26: It is recommended that the phrase “results from” in the title be omitted, as this would shorten the title without detracting from its intended meaning.

Thank you for this suggestion – we have removed “results from” the title and agree that the shortened version sounds better.

Pg. 2/26: - It is recommended that the phrase “and Outcomes” be included in the title, as currently presented at the end of the background, to ensure that the title is complete.

We believe these titles are given by the journal and thus are not allowed to modify them. We do state the objective in the background; the outcomes are provided in the Methods subsection.

- Please ensure that the sampling method, data collection tools, a brief description of the data collection procedure, and the statistical tests used are clearly described in the methods section.

We have expanded the Methods section to provide more details: it now reads as follows:

“We conducted a 2-arm open label randomized controlled trial involving 6,000 randomly selected women, ages 18-49 years, with access to a simple feature Airtel cell phone in Uganda. Fifty enumeration areas were randomly selected in Kampala, Madi-Okollo, Rwampara and Katakwi districts. Consenting women were asked to complete a baseline and endline survey implemented by trained interviewers using tablets through home visits. After completion of baseline, women were randomized with equal probability to treatment and control using a tablet-generated random number draw. Study staff introduced treated women to the call-in service and encouraged its use through various incentives and promotional messages. Primary outcomes were sexual and reproductive health knowledge scores as well as iron supplementation in pregnancy. Secondary outcomes included self-reported use of contraception, healthy diets, birth spacing and birth outcomes as well as hemoglobin levels and BMI. Linear regression models were used to estimate mean differences in outcomes between treatment and control. Standard errors were corrected for the cluster-based sampling of women.”

Pg. 3/26: Please mention keywords at the end of the abstract.

We have added the following keywords at the end of the abstract:

Keywords: reproductive health, call-in-services, health knowledge, Uganda, Viamo.

Pg. 4/26: Please provide further details regarding the Viamo platform and its operational mechanisms. Specifically, clarify whether the platform enables two-way interaction and whether users can experience a form of interactive support comparable to counseling, or whether information is delivered in a predominantly one-way manner, functioning mainly as a communication channel for educational purposes.

Please review relevant studies that have implemented similar interventions in the same research area, either locally or in other regions. Additionally, address studies that have evaluated the effectiveness of the Viamo platform or comparable mobile- or voice-based platforms for different purposes. Finally, summarize the existing literature and highlight the comparative advantage and necessity of the present intervention by identifying the gaps it aims to address. If no relevant studies are available, this should be clearly stated.

Based on the feedback provided by Reviewer 4, we have removed the description of the service from the Introduction (page 4) and instead provide more details on the platform in an expanded Methods-Intervention section, where we provide the requested details. We compare this to other studies in the Discussion section – there is however not that much literature on similar models. In practice, Viamo does not provide direct counseling, but it offers a range of different topics users can select from – we think it can be best described as something like a “user-guided learning system”.

Pg. 5/26 (Methods section):

- Please specify the exact nature of any encouragement designs and whether they may have preferentially attracted certain subgroups of participants. For example, as noted on page 7, "Participants received UGX 10,000 (~USD $2.77) of phone credit as a token of gratitude after each survey interview." Offering such incentives could disproportionately attract individuals with lower socioeconomic status, potentially limiting

---

## [Editor Report · Decision Letter 2]

22 Jan 2026

Estimating the Impact of Viamo Call-in Information Services on Sexual and Reproductive Health Knowledge, Behavior, and Outcomes among Women of Reproductive Age: A 2-Arm Open Label Randomized Controlled Trial in Uganda

PONE-D-25-05733R2

Dear Dr. Fink,

We’re pleased to inform you that your manuscript has been judged scientifically suitable for publication and will be formally accepted for publication once it meets all outstanding technical requirements.

Kind regards,

Alfredo Luis Fort, M.D., M.Sc., Ph.D.

Academic Editor

PLOS One

PS: You have made changes and adaptations to the article according to previous reviewers. So, we are now accepting the publication of the manuscript (and I am including a few very minor notes in the attached file for suggested mini-changes but which can also be contemplated by the publishing editor at the time of pre-publication). Best.

---

## [Editor Report · Acceptance letter]

PONE-D-25-05733R2

PLOS One

Dear Dr. Fink,

I'm pleased to inform you that your manuscript has been deemed suitable for publication in PLOS One. Congratulations! Your manuscript is now being handed over to our production team.

Kind regards,

on behalf of

Dr. Alfredo Luis Fort

Academic Editor

PLOS One